# Excessive C5 conversion prevents C9 polymerisation and subsequent MAC-dependent killing of *Klebsiella pneumoniae*

Kulsum M. Dawoodbhoy[1], Coco R. Beudeker[1,2], Panagiotis Theofilidis[1], Frerich M. Masson[1], Michiel van der Flier[1,2], Suzan H. M. Rooijakkers[1], Bart W. Bardoel[1*], Dennis J. Doorduijn[1]

1 Department of Medical Microbiology, University Medical Center Utrecht, Utrecht, The Netherlands,
2 Wilhemina Children's Hospital, Pediatric Infectious Diseases and Immunology, University Medical Centre Utrecht, Utrecht, The Netherlands

* b.w.bardoel-2@umcutrecht.nl

## Abstract

Membrane Attack Complex (MAC) pores are important in the human innate immune response to directly kill pathogenic Gram-negative bacteria. MAC pores assemble when complement proteins in serum are activated on bacteria and convert complement protein C5 into C5b, which together with C6, C7, C8, and multiple copies of C9 form a pore that damages the bacterial envelope. Due to rising multidrug-resistant infections with Gram-negative pathogen *Klebsiella pneumoniae* (Kpn), there is interest in developing complement-activating monoclonal antibodies (mAbs) that trigger MAC-dependent killing. However, some Kpn strains resist MAC-dependent killing in serum despite potent complement activation and C5 conversion, revealing a critical gap in understanding how these strains resist MAC-dependent killing. We demonstrate that Kpn strains can resist MAC-dependent killing through a paradoxical mechanism of excessively converting C5, which limits C9 polymerisation and subsequent killing. In these strains, spiking serum with supplementary C9 restored killing. Additionally, partially inhibiting C5 conversion using complement inhibitors increased C9 polymerisation and subsequent killing. This suggests that these Kpn strains are in principle sensitive to MAC-dependent killing, but an imbalance between generated C5b and available C9 in serum limits C9 polymerisation and prevents killing. We also observed this paradoxical effect with an of excess complement-activating mAbs on Kpn strains that are typically susceptible to MAC-dependent killing in serum. Excessive C5 conversion was responsible for this reduced killing, as supplementary C9 restored killing. Lastly, in neonatal plasma, where C9 is naturally limited, complement-activating mAbs induced killing of Kpn only in the presence of supplementary C9. Our study highlights that the balance between C5 conversion and available C9 is important for MAC-dependent killing of Kpn. Additionally, absence of killing in serum does not necessarily indicate that Kpn strains are MAC-resistant. These insights are important

**Data availability statement:** All revelant data are in the manuscript and supplementary data file. This paper does not report original code. Klebsiella pneumoniae whole-genome sequences are available on figshare.com (DOI:10.6084/m9.figshare.31971786).

**Funding:** This work was funded by the European Research Council (ERC) under the European Union's Horizon 2020 research and innovation program (grant agreement no. 101001937, ERC-ACCENT, to S.H.M.R., https://erc.europa.eu/homepage) and Wilhelmina Children's Hospital Fund (Grant 2019-2020, to MvdF, https://www.hetwkz.nl/nl). The funders had no role in study design, data collection and interpretation, or the decision to submit the work for publication.

**Competing interests:** The authors have declared that no competing interests exist.

for interpreting mAb efficacy in serum bactericidal assays and in complement-deficient populations.

## Author summary

Antibiotic-resistant infections caused by *Klebsiella pneumoniae* represent a major global health challenge. The human immune system normally combats such bacteria by activation of complement proteins, a group of molecules in the blood that can assemble the Membrane Attack Complex (MAC). The MAC forms pores that puncture the bacterial cell envelope and kill bacteria. Here, we found that potent activation of complement proteins can paraxodically prevent killing of certain *K. pneumoniae* strains. Excessive conversion of MAC component C5 limits the assembly of MAC pores consisting of multiple copies of C9, which are required for killing. We show that when extra C9 was added, the bacteria were killed, demonstrating that they are not truly resistant to MAC pores but escape through this imbalance. We also observed this effect in newborn samples, where natural C9 levels are lower and when testing antibody-based therapies designed to boost complement activity. These findings reveal that the absence of bacterial killing in laboratory assays does not always reflect genuine resistance, but can result from disproportional activation. Recognising this mechanism is critical for developing effective antibody therapies, protecting vulnerable patients, and addressing the growing threat of antibiotic-resistant infections.

## Introduction

The complement system is a key component of innate immunity that directly kills bacterial pathogens. Through a proteolytic cascade, complement activation ultimately results in the formation of the membrane attack complex (MAC) pores [1]. Antibodies can initiate complement activation by binding to C1, leading to the sequential assembly of C3 convertases, resulting in the deposition of C3b, and C5 convertases [2]. These C5 convertases generate C5b, which sequentially binds C6, C7, C8, and multiple copies of C9 that polymerise to form a MAC pore [3–7]. Once assembled, these pores damage the bacterial cell envelope ultimately causing bacterial killing of Gram-negative bacteria [5].

*Klebsiella pneumoniae* (Kpn) is one of the key Gram-negative pathogens of concern that is a major cause of hospital-acquired infections, particularly in immunocompromised individuals and neonates [8,9]. The rise in prevalence of multidrug-resistant (MDR) Kpn strains has made treatment of infection with Kpn increasingly difficult, highlighting the need for alternative therapeutic strategies [10]. Kpn-targeted monoclonal antibodies (mAbs) have emerged as a promising alternative, as these can trigger complement activation to promote opsonophagocytosis and drive MAC formation to directly kill Kpn [11,12].

However, Kpn has evolved multiple strategies to resist MAC-dependent killing [13,14]. A recent report identified a Kpn O1-antigen strain that resisted MAC-dependent killing despite potently converting C5, which was associated with decreased C9 polymerisation and incomplete, improperly inserted MAC pores that were released as soluble MAC (sMAC) [15]. This phenomenon has previously also been observed in *E. coli* [5,16]. These findings raise critical questions about the relationship between C5 conversion and C9 polymerisation, in particular whether an excess of C5b can limit the formation of polymerised C9 and subsequent bactericidal MAC pores.

Answering these questions is especially relevant when considering mAb therapy focused on boosting complement activation on bacteria. Serum bactericidal assays, which are frequently used to assess MAC-dependent killing by mAbs *in vitro* have been described to display a 'hook-curve', where increased antibody binding paradoxically reduces bacterial killing [17]. Understanding this balance is also important to determine the potential of mAbs to induce MAC-dependent killing in individuals with complement deficiencies. For instance, neonates have naturally low levels of complement proteins, including MAC component C9, due to their immature immune system, which could prevent efficient MAC pore formation [18,19].

In this study, we demonstrate that supplementing serum with C9 or partially inhibiting C5 conversion increased C9 polymerisation and restored MAC-dependent killing in Kpn strains that potently convert C5. This suggests that excessive C5 conversion can limit C9 polymerisation and subsequent MAC-dependent killing. Additionally, we found that potent C5 conversion triggered by an excess of Kpn-targeting antibodies can reduce MAC-dependent killing of an otherwise serum-sensitive strain. Supplementing with C9 restored killing in this context, suggesting that this reduction in MAC-dependent killing is also the result of excessive C5 conversion. Lastly, in neonatal plasma, we demonstrate that C9 supplementation is necessary for antibody-dependent MAC-dependent killing of Kpn strains. These findings provide mechanistic insights into how some Kpn strains resist MAC-dependent killing through a paradoxical mechanism of excessively converting C5. This has important implications for the interpretation of serum bactericidal assays, and development of complement-targeted interventions in bacterial infections.

## Results

### Supplementing C9 in human serum increases killing of a Kpn strain that resists MAC-dependent killing despite potently converting C5

We started by testing our hypothesis of whether the available C9 in NHS (normal human serum) limits MAC-dependent killing when C5 is potently converted. In NHS, the molar concentration of C9 (~900 nM [20]) exceeds that of C5 (~375 nM [21]), resulting in a ratio of approximately 2–3 C9 molecules per C5. This is insufficient for forming a fully assembled MAC pore, previously described as containing 18 polymerised C9 monomers [5,22].

We examined whether supplementing serum with exogenous C9 could restore MAC-dependent killing in a Kpn O1-antigen strain, KpO1_1, which potently converts C5 in the absence of killing [15]. Bacteria were incubated in 5% NHS, and inner membrane- (IM) permeabilisation was measured by the influx of the membrane impermeable DNA dye, Sytox, over time. This assay is a sensitive and established read-out for killing [6,23]. We observed minimal IM-permeabilisation over 90 minutes with NHS alone (Fig 1a). However, supplementing 560 nM exogenous C9 (± equivalent to 66.6% serum) rapidly permeabilised the IM within 30 mins (Fig 1a). Counting colony-forming units (CFUs) by plating confirmed that supplementing C9 significantly reduced bacterial viability by a factor of 10 compared to NHS alone (Fig 1b). Administration of C5 conversion inhibitors (OmCI and Eculizumab) abrogated this killing effect (Fig 1a-b), confirming that killing of KpO1_1 is MAC-dependent. MAC-dependent damage of KpO1_1 increased in a dose-dependent manner when C9 concentration exceeded C9 level at 5% serum (42 nM) (Fig 1b-c), further indicating that the available C9 in NHS is limiting for MAC-dependent killing of KpO1_1. In sum, these findings indicate that supplementing C9 can increase MAC-dependent killing of a Kpn strain that potently converts C5.

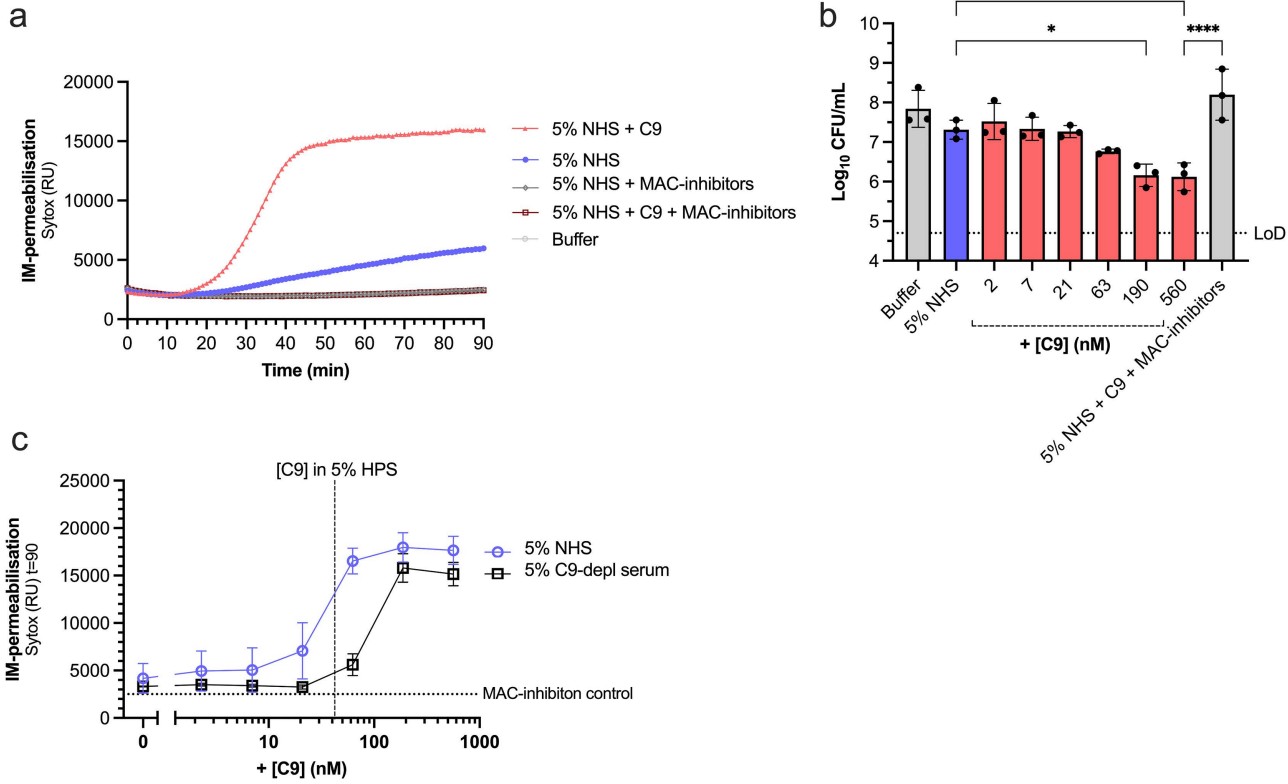

**Fig 1. Supplementing C9 in NHS increases MAC-dependent killing of a Kpn strain.** Kpn KpO1_1 (~5 × 10⁷ bacteria/ml) was incubated with 5% NHS or supplemented with 560 nM C9 and/or MAC-inhibitors (6 µg/ml OmCI and 15 µg/ml Eculizumab) at 37 °C for 90 min. SYTOX fluorescence intensity was detected as read-out for IM-permeabilisation every min in a microplate reader **(a)**. **(b-c)** KpO1_1 was incubated in 5% NHS or 5% C9-depleted serum (b) with a concentration range of C9 for 90 min at 37 °C. Bacterial viability is depicted as log-transformed CFU/ml values **(b)**. The horizontal dotted line represents the assay's limit of detection (LoD). SYTOX fluorescence intensity after 90 min was plotted as read-out for IM-permeabilisation **(c)**. Dotted horizontal line indicates MAC-inhibition control (5% NHS with MAC-inhibitors) and dotted vertical line indicates the C9 concentration in 5% NHS. **(a)** Representative IM-permeabilisation assay from three independent experiments. **(b,c)** Data are from three independent experiments. Statistical analysis was performed for **(b)** using a one-way ANOVA with Tukey's multiple comparisons' test. Only significant conditions are indicated as *p < 0.05 and ****p<0.0001.

## Supplementing C9 increases MAC-dependent killing in multiple Kpn strains that potently convert C5

We next investigated whether supplementing C9 could restore MAC-dependent killing for more Kpn clinical isolates, selected to represent a diverse range of O- and K-antigen types (Table 1) [15,25]. We screened a panel of eight Kpn strains and assessed for IM-permeabilisation and MAC-dependent killing (Fig 2a-b, S1a-h Fig). Reduction in bacterial viability attributable to supplementary C9 was quantified as the fold change in CFU/ml between NHS and NHS with supplementary C9 for each Kpn strain (Fig 2c). In addition to strain KpO1_1 (Fig 1), supplementing C9 also increased bacterial killing by >1-log and increased IM-permeabilization for KpO3_4 (Fig 2a-c, S1b Fig). Five other strains (Kp209, KpO2_3, KpO5_1, KpO3_3 and KpO5_2) displayed a similar trend in IM-permeabilisation and bacterial killing attributable to supplementary C9, although bacterial killing increased below <1-log (Fig 2a-c, S1c-h Fig). Supplementing C9 at higher serum concentrations above 5% also increased bacterial killing by >1-log (S2a) and increased IM-permeabilisation increased over multiple timepoints (S2b-c). Without supplementary C9, KpO2_1 was already killed, suggesting a higher sensitivity to MAC-dependent killing (Fig 2a, b, S1h Fig). Altogether, these data suggest that the C9 concentration limits

**Table 1. Bacterial strains used in this study. Kpn strain nomenclature follows previous publications. Strains are named according to their O-antigen type, except for Kp209 and Kp209_CSTR.**

| Strain | O type | K type | Origin |
|---|---|---|---|
| KpO1_1 | O1 | KL114 | Masson et al., 2024 [15] |
| KpO2_1 | O2a | KL114 | Medical Microbiology Department, University Medical Centre Utrecht |
| KpO2_3 | O2afg | KL102 | Masson et al., 2024 [15] |
| Kp209 | O2 | KL110 | Janssen et al., 2020 [24] |
| Kp209_CSTR | O2 | KL110 | Janssen et al., 2020 & Van der Lans et al., 2023 [25,24] |
| KpO3_3 | O3b | KL125 | Van den Bunt et al., 2020 [26] |
| KpO3_4 | O3b | No confidence match | Van den Bunt et al., 2020 [26] |
| KpO5_1 | O5 | No confidence match | Van den Bunt et al., 2020 [26] |
| KpO5_2 | O5 | KL10 | Van den Bunt et al., 2020 [26] |

MAC-dependent killing in NHS of multiple Kpn strains. As these strains are of different O-antigen types (Table 1), it also suggests that this is not restricted to a single O-antigen type.

Next, we wanted to assess if the additional killing attributable to C9 is linked to potent C5 conversion. Since the ratio of C5:C9 in NHS is about 1:3 [20,21], potent C5 conversion could limit the amount of C9 molecules available to assemble complete MAC pores. C5 conversion was quantified using an ELISA to detect the released C5a into the supernatant as an estimate for the generation of MAC precursor C5b [27]. Approximately all C5 present in 5% NHS (~20 nM of C5) was converted for KpO1_1 and KpO3_4 (Fig 2d), as opposed to ~2.5-7.5 nM for the other Kpn strains. This suggests the pronounced increase in bacterial killing by supplementary C9 in NHS for KpO1_1 and KpO3_4 could be linked to potent C5 conversion (Fig 2a, c). KpO2_2 generated the lowest C5a levels (Fig 2d). Flow cytometry using an anti-C3b antibody revealed reduced C3b deposition on KpO2_2 compared to other Kpn strains, suggesting less efficient upstream complement activation, which prevents efficient C5 conversion and MAC-dependent killing (Fig 2e, S2c Fig).

Lastly, we found that C5 conversion levels positively correlates ($R^2 = 0.6$) with reduction in bacterial viability attributable to supplementary C9, with a clear separation between KpO1_1 and KpO3_4 and the rest of the strains (Fig 2f). Altogether, these data not only demonstrate that C9 supplementation can enhance MAC-dependent killing for Kpn strains with potent C5 conversion, but also indicate that the available C9 in serum can be limiting under conditions of potent C5 conversion.

## Potent C5 conversion can limit C9 polymerisation and subsequent MAC-dependent killing

We next wanted to more directly test if potent C5 conversion can prevent MAC-dependent killing by limiting C9 polymerisation. Potent C5 conversion could result in an excess of C5b-8 MAC precursors, which has been shown to limit C9 polymerisation [5]. Importantly, C9 polymerisation is crucial for efficient MAC-dependent killing of Kpn [5].

To test this hypothesis, C5 was titrated in 5% C5-depleted serum to determine if an excess of C5b can decrease bacterial killing of KpO1_1, which is hereafter used as a model strain for potent C5 conversion. Although increasing amounts of C5 initially increase bacterial killing, killing sharply declines at concentrations exceeding >2 times-the C5 concentration in 5% serum (Fig 3a). This suggests that an excess of C5b can prevent MAC-dependent killing in serum. Conversely, C5 conversion inhibitor, OmCI, was titrated into serum to gradually decrease the formation of C5b molecules leading to an increase in the C9:C5b ratio. OmCI concentrations above 0.07 µg/ml enhanced MAC-dependent killing of KpO1_1 up to 2-log fold in 5% NHS (Fig 3b, S3a Fig), which coincided with increased IM-permeabilisation (S3b Fig). An ELISA to assess C5a release in the assay supernatant confirms that at 0.07 µg/ml of OmCI, C5 conversion is partially inhibited (Fig 3c). This suggests that enough C5b is generated to assemble MAC pores to kill bacteria. At concentrations of OmCI exceeding

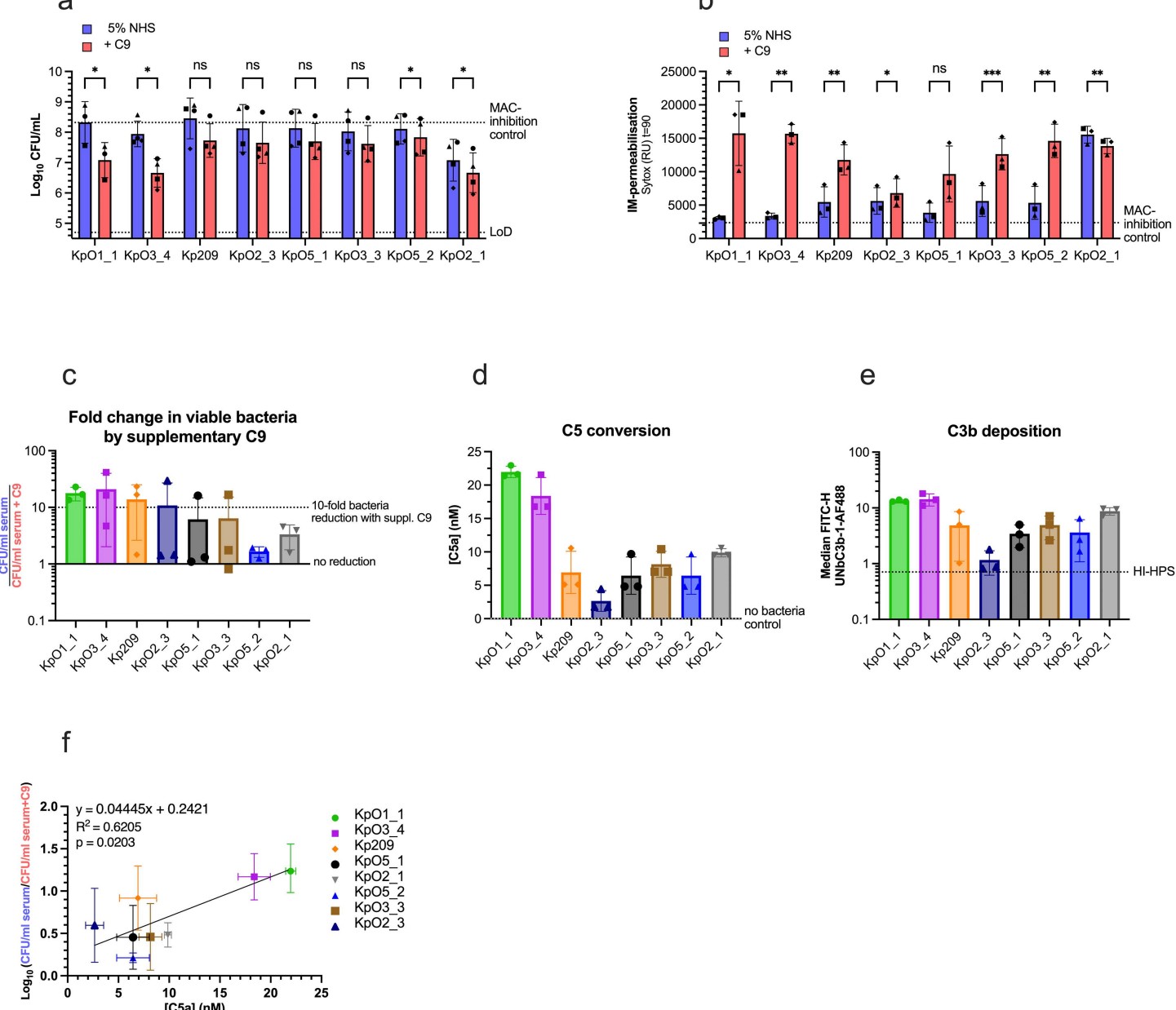

**Fig 2. Supplementing C9 into NHS increases MAC-dependent killing for multiple Kpn strains that potently convert C5. (a)** Bacterial viability and **(b)** IM-permeabilisation as determined by SYTOX fluorescence for multiple Kpn strains after 90 min. Multiple Kpn strains (each ~5 × 10$^7$ bacteria/ml) were incubated with 5% NHS or 5% NHS supplemented with 560 nM C9 at 37°C for 90 min. Bacterial viability is depicted as log-transformed CFU/ml values. The upper horizontal dotted line depicts the mean MAC-inhibition control (5% NHS supplemented 6 µg/ml OmCI and 15 µg/ml Eculizumab) across all Kpn strains and the lower horizontal dotted line represents the assay's limit of detection (LoD). Each symbol shape represents an experimental repeat. **(c)** Reduction in bacterial viability attributable to supplementary C9 was calculated from values in (a) as the fold change of the CFU/ml between the 5% NHS alone and 5% NHS supplemented with 560 nM C9. **(d,f)** Kpn strains were incubated in 5% NHS at 37°C for 90 min. **(d)** Supernatant was collected and C5a was quantified with a sandwich ELISA. Horizontal dotted line indicates the no bacterial control (5% NHS only) value. **(e)** C3b deposition was detected using UNbC3b-1-AF488 and analysed by flow cytometry using median fluorescence intensity of the bacterial population. **(f)** Correlation between C5 conversion (measured by C5a production; shown in **(e)**) plotted on the x-axis and reduction in bacterial viability attributable to supplementary C9 (shown in **(c)**) plotted on the y-axis. The line of best fit and R$^2$ value were determined by a linear regression analysis. Data represents mean ± SD of three **(b, c, e)** or four **(a)** independent experiments (except for the KpO1_1 condition, where three independent experiments were conducted). **(f)** Data represents mean and ± SEM from three independent experiments. **(d)** Data represents one individual experiment using supernatants from three independent experiments. Statistical analysis was performed for **(a, b)** using multiple paired t-tests; significant differences are displayed as *p < 0.05 and **p < 0.01.

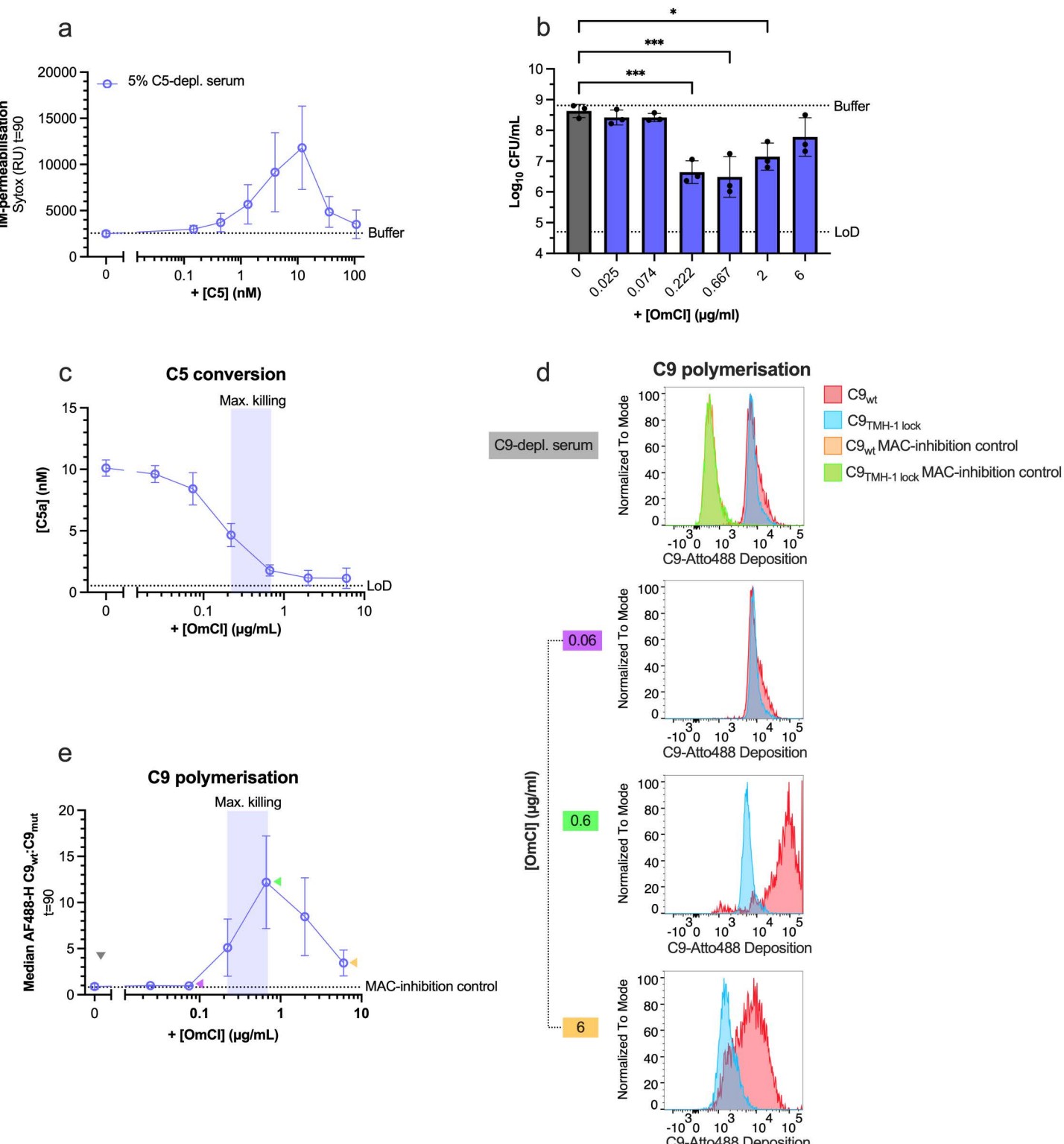

**Fig 3. Potent C5 conversion can limit polymerisation and subsequent MAC-dependent killing. (a)** KpO1_1 (~5 × 10⁷ bacteria/ml) was incubated with 5% C5-depleted serum supplemented with a concentration range of C5 at 37 °C and SYTOX fluorescence intensity was measured after 90 min. **(b)** KpO1_1 was incubated in 5% C9-depleted serum repleted with 5% serum equivalent of C9-Atto488 (40 nM) and a concentration range of OmCI (C5 inhibitor) at 37 °C for 90 mins. Bacterial viability is depicted as log-transformed CFU/ml values. **(c)** KpO1_1 was incubated in 5% NHS at 37 °C

for 90 min. Supernatant was collected and C5a was quantified by a sandwich ELISA, as a readout for C5 conversion. **(d, e)** KpO1_1 was incubated for 90 min at 37 °C in 5% C9-depleted serum or repleted with 5% equivalent of either C9$_{wt}$-Atto488 or C9$_{TMH-1\ lock}$-Atto488 with a concentration range of OmCI. Atto488 fluorescence was detected by flow cytometry. **(d)** C9$_{wt}$ and C9$_{TMH-1}$ deposition on KpO1_1 at each OmCI concentration is depicted by representative histograms. **(e)** C9 polymerisation index was calculated by dividing the median of the bacterial population incubated with C9$_{wt}$-Atto488 by the median of the bacterial population incubated with C9$_{TMH1-lock}$-Atto488. Where present, shaded regions indicate OmCI concentrations that led to maximum killing. Horizonal dotted lines represent the assay's limit of detection (LoD), buffer control or MAC-inhibition control (C9-depleted serum supplemented with 6 µg/ml OmCI and 15 µg/ml Eculizumab), as indicated. **(a, b, e)** Data represents mean ± SD of three independent experiments. **(c)** Data represents one individual experiment using supernatants from three independent experiments. **(d)** Representative histogram from three independent experiments. Statistical analysis was performed for **(b)** using a one-way ANOVA with Tukey's multiple comparisons' test. Only significant differences versus 0 µg/ml OmCI condition are indicated as *p < 0.05 and ***p ≤ 0.001.

2 µg/ml C5a levels drop below detectable levels, suggesting effective inhibition of C5 conversion (Fig 3c), which corresponds with the absence of MAC-dependent killing (Fig 3b). Titrating eculizumab, another C5 conversion inhibitor into serum was consistent with the effects of OmCI (S4a Fig). These data suggest that partial inhibition of C5 can enhance MAC-dependent killing of KpO1_1.

To verify if high levels of C5b inhibit C9 polymerisation and thus MAC-dependent killing, we measured C9 polymerisation on the bacterial surface. KpO1_1 was incubated with 5% C9-depleted serum, a concentration range of OmCI and either fluorescently labelled wild-type C9 (C9$_{wt}$) or a non-polymerizing mutant C9 (C9$_{TMH-1\ lock}$) that can bind C5b-8 but cannot recruit additional C9 molecules to assemble a complete MAC pore [5,28]. A single fluorophore was site-specifically attached to both C9$_{wt}$ and C9$_{TMH-1\ lock}$ using sortase-mediated labelling, ensuring a one-to one ratio of fluorophore to C9 molecule. Therefore, the difference in binding between C9$_{wt}$ and C9$_{TMH-1\ lock}$ determined by flow cytometry was used as a read-out for C9 polymerisation, as used previously [5,15] (Fig 3d). As expected, this C9$_{TMH-1\ lock}$ resulted in less efficient bacterial killing than C9$_{wt}$ (S3c Fig). Increasing OmCI concentrations till 0.7 µg/ml increased polymerisation of C9 to a maximum of 12 C9 molecules, which coincides with the peak of MAC-dependent killing (Fig 3d). C9 polymerisation decreased above 0.7 µg/ml OmCI, aligning with the absence of bacterial killing (Fig 3b, e). Since C9 polymerisation does not return to the level observed in the condition without OmCI, some additional bacterial killing might be expected. However, the total deposition of both C9$_{wt}$ and C9$_{TMH-1\ lock}$ was substantially reduced relative to the condition without OmCI (Fig 3d), corresponding with undetectable C5a levels (Fig 3c). This suggests that there were too few MAC pores for bacterial killing.

Furthermore, we show the effects of OmCI and eculizumab are consistent with other complement inhibitors. Nanobody UNbC3b-1, which inhibits C5 conversion via the alternative pathway, also resulted in a bell-shaped IM-permeabilisation curve with a peak at 1.6 µg/ml (S4b Fig). This coincided with maximum killing showing up to 1-log fold increase in killing compared to the without UNbC3b-1 condition and enhanced C9 polymerisation (S4b-d Fig). Similarly, titration of classical pathway inhibitor C1qNb75 [29], which prevents C1q from binding to antibodies, increased IM-permeabilisation of KpO1_1 (S4e Fig). These data further support that partial inhibition of C5 conversion increases MAC-dependent killing independent of the complement pathway.

In sum, these findings indicate that potent C5 conversion can limit C9 polymerisation and subsequent MAC-dependent killing of Kpn in human serum.

## An excess of Kpn-specific antibodies can decrease MAC-dependent killing of Kpn by potent C5 conversion

We next wondered if an excess of Kpn-specific antibodies can trigger potent C5 conversion and inhibit MAC-dependent killing of an otherwise serum-sensitive Kpn. We developed a model system that allowed us to control antibody-dependent complement activation with Kpn-specific mAbs and serum preadsorbed with Kpn strains (dNHS) to deplete pre-existing anti-Kpn antibodies [25].

In this system, a complement-activating IgG1 monoclonal antibody against the O2-antigen of KpO2_1 (UKpn2) [17] dose-dependently increased IM-permeabilisation (Fig 4a), and bacterial killing up to 10-fold (Fig 4d). However, at concentrations above 0.3 µg/ml, both IM-permeabilisation and bacterial killing returned to similar levels of serum without antibody

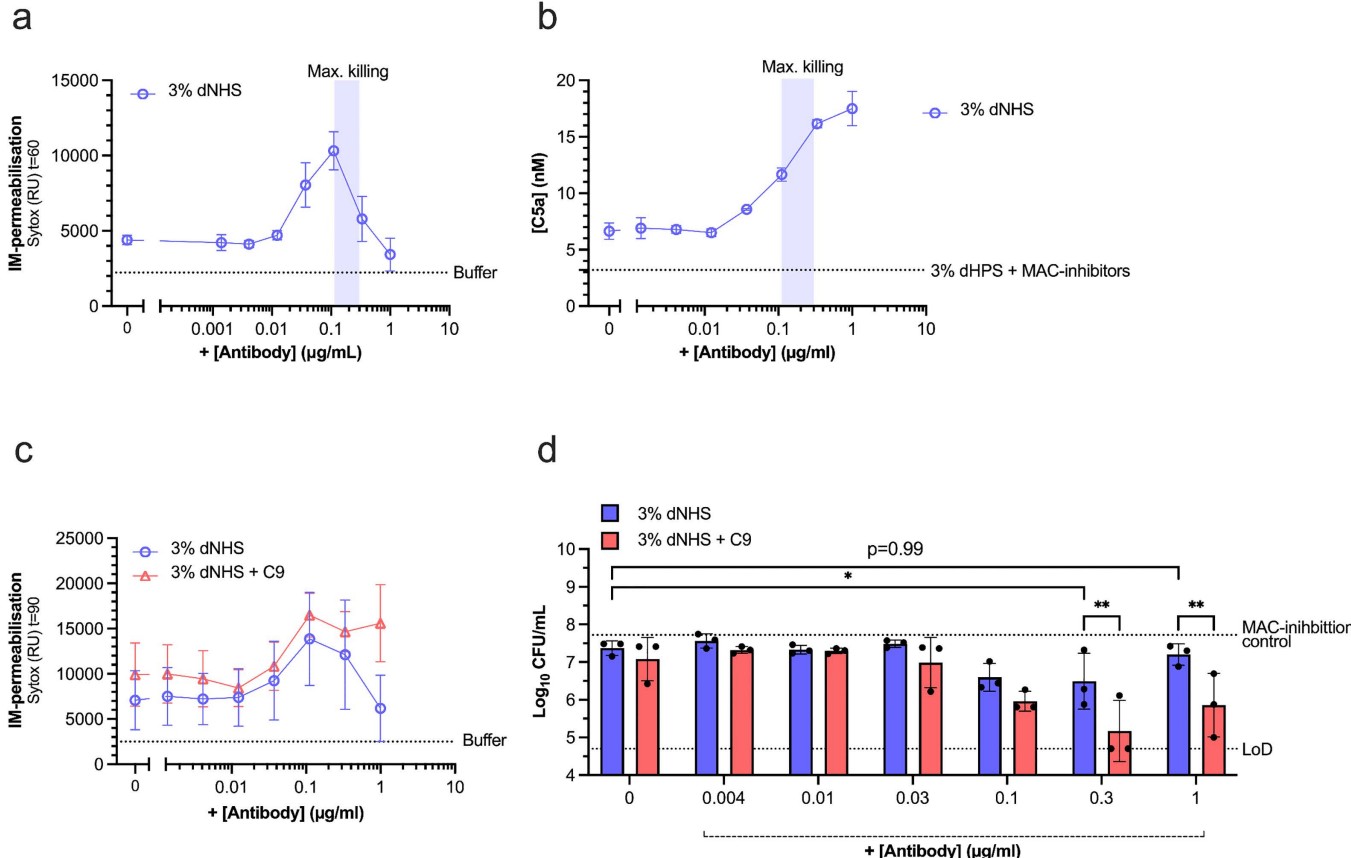

**Fig 4. An excess of Kpn-specific antibodies can decrease MAC-dependent killing of Kpn by potent C5 conversion. (a, b, c, d)** Kpn SF002 (~5 × 10⁷ bacteria/ml) was incubated in 3% NHS preabsorbed with Kpn strains to deplete pre-existing anti-Kpn antibodies (dNHS), as a complement source, with a concentration range of UKpn2 antibody (anti-Kpn O2-ag IgG1) at 37 °C for 90 min. For **(c, d)**, 3% dNHS was additionally supplemented with 560 nM C9. IM-permeabilisation was depicted by plotting SYTOX fluorescence intensity after **(a)** 60 min or **(c)** 90 min. **(b)** C5 conversion was determined using reaction supernatant after 90 min to quantify C5a by a sandwich ELISA. **(d)** Bacterial viability was determined after 90 min and depicted as log-transformed CFU/ml values. Where present, shaded regions indicate UKpn2 antibody concentration that led to maximum killing. Horizontal dotted lines represent the level of the buffer control, MAC-inhibition control (3% dNHS with 6 µg/ml OmCI and 15µg/ml Eculizumab) or the assay's limit of detection (LoD), as indicated. Data represents mean ± SD of three **(a, d)** or **(c)** five independent experiments. **(b)** Data represents one individual experiment using supernatants from three independent experiments. **(d)** Statistical analysis was performed between 3% dNHS with or without supplementary C9 conditions using a two-way ANOVA with Šídák's multiple comparisons' test. Additionally, a one-way ANOVA's with Tukey's multiple comparisons test was conducted to compare all 3% dNHS conditions without supplementary C9 to the 0 µg/ml-antibody condition. Significant differences are indicated as *$p < 0.05$ and **$p ≤ 0.01$.

(Fig 4a, d), revealing a 'hook-curve', as mentioned previously [17]. By contrast, C5 conversion did further increase above 0.1 µg/ml UKpn2, suggesting that potent C5 conversion by antibodies coincided with a decrease in bacterial killing (Fig 4b). Supplementing C9 in these conditions restored IM-permeabilisation and decreased viability at UKpn2 concentrations above 0.1 µg/ml, further suggesting that the potent C5 conversion reduced killing because of the limited C9 available (Fig 4c-d). Together, these results demonstrate that an excess of antibodies can drive potent C5 conversion which prevents MAC-mediated killing of a serum-sensitive Kpn strain.

**C9 concentration in neonatal plasma is limiting MAC-dependent killing of *Klebsiella pneumoniae* by antibodies**

Finally, we wondered if killing of MAC-susceptible Kpn strains would be affected under physiologically relevant conditions of limited C9, as observed in plasma from neonates, with levels of 24 mg/l [18,19] corresponding to ~40% of adult levels at 60 mg/l.

KpO2_1 and Kp209_CSTR were each incubated in 5% full neonatal plasma with or without supplemental C9 that is equivalent to 25% adult serum. Since we expect that neonatal plasma contains few Kpn-specific antibodies [30,31], we added UKpn2 IgG1 to initiate complement activation. No IM-permeabilisation was observed on KpO2_1 or Kp209_CSTR when incubated with neonatal plasma alone from 8 donors (Fig 5a-b). Supplementing the neonatal plasma with UKpn2 or C9 alone did not increase IM-permeabilisation (Fig 5a-b). Only when UKpn2 and C9 were co-administered together, extensive IM-permeabilisation was observed up to the level of adult serum (Fig 5a-b). Administration of C5 conversion inhibitors OmCI and Eculizumab abrogated this effect (Fig 5a-b), confirming that killing of KpO2_1 and Kp209_CSTR is MAC-dependent. Altogether, these findings demonstrate the capacity of mAbs to induce MAC-dependent killing of Kpn in neonatal plasma is impaired because of a combination of low antibody titres against Kpn and limited availability of C9.

## Discussion

Traditionally, increased complement activation is thought to enhance MAC formation and subsequent bacterial killing. However, recent studies have revealed that some Gram-negative bacteria can resist MAC-dependent killing in human

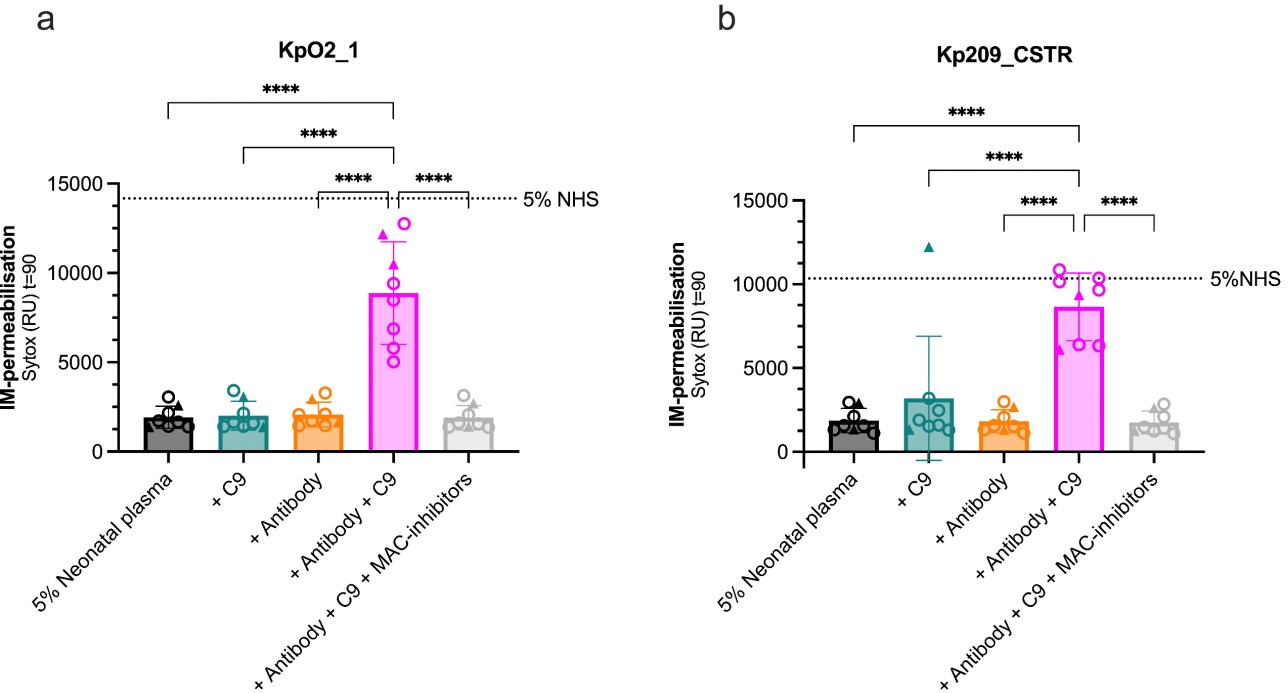

**Fig 5. In neonatal plasma, supplementing a Kpn-specific antibody plus C9 increases MAC-dependent killing of serum-sensitive Kpn strains in neonatal plasma. (a)** Kpn SF002 or **(b)** 209R (each ~5 × 10⁷ bacteria/ml) were incubated in 5% neonatal plasma supplemented with either 210 nM C9 and/or 1 µg/ml UKpn2 (anti-Kpn O2-ag IgG1), or with both plus MAC-inhibitors (6 µg/ml OmCI and 15 µg/ml Eculizumab) at 37 °C. IM-permeabilisation was depicted by plotting SYTOX fluorescence intensity after 90 min. Individual symbols represent different neonatal plasma donors (n = 8), where ○ indicates preterm and △ indicates term neonates. Horizontal dotted line represents the level of 5% NHS. Data represents mean ± SD. Data is representative of two independent experiments. Statistical analyses were performed using a one-way ANOVA with Tukey multiple comparison analysis. Significant differences are indicated as ****p < 0.0001.

serum despite potent C5 conversion [5,15,16]. It remained unclear how the intensity of C5 conversion affects the assembly of C9 polymers to form MAC pores that can kill bacteria. Our study addresses this gap and demonstrates that potent C5 conversion paradoxically limits C9 polymerisation, which reduces MAC-dependent killing of multiple Kpn strains. We also show that an excess of complement-activating mAbs can decrease MAC-dependent killing of Kpn through the same mechanism. Together, these findings suggest that excessive C5 conversion can prevent MAC-dependent killing.

Either supplementing C9 in serum or decreasing C5 conversion with complement inhibitors increased C9 polymerisation and subsequent MAC-dependent killing of Kpn that potently convert C5. This suggests that an imbalance in the generated C5b-to-C9 ratio limits C9 polymerisation and MAC-dependent killing. A previous study that incubated *E. coli* with C8-deficient serum, washed the serum away, and then supplemented with C8 and varying concentrations of C9 also showed that an excess of C9 compared to C5b-8 is required for optimal killing of *E. coli* [32]. We extend on these findings by demonstrating that C9 can be limiting for MAC-dependent killing of Kpn in full human serum. This highlights the relevance of this mechanism under physiological settings and shows that the absence of bacterial killing in serum does not necessarily indicate MAC-resistance.

Our data indicate that maximum bacterial killing coincides with the highest observed C9 polymerisation, which indicates approximately 1 C5b to 12 C9 molecules. This is consistent with previous EM and biochemical studies suggesting a ratio of 1 C5b to 12–18 C9 molecules, and are on the lower end of cryo-EM structures demonstrating a C5b:C9 ratio of 1:22 [33,34]. Moreover, our findings corroborate data in *E. coli* J5, where ≥3 C9 molecules per C5b-8 are required for killing and a C9:C5b-8 ratio of 11:1 achieves >99.9% killing [32].

We hypothesise that under limiting C9 conditions, multiple C5b-8 complexes may capture single C9 molecules without progressing to full pore formation. Thus, when C9 is supplemented or when C5 conversion is partially inhibited, the ratio of C5b-to-C9 favours assembly of C9 polymers, allowing for rapid oligomerisation into bactericidal MAC pores. These findings align with our previous findings that C9 binding to C5b-8 is comparable to, or even favoured over, binding to polymeric-C9 in the nascent C5b-9 complex [5]. In contrast, another study using atomic force microscopy experiments on supported lipid bilayers derived from bacterial membrane extracts observed binding of C9 to C5b-8 is the rate-limiting step of the assembly of C9 polymers [35]. Future studies applying direct biochemical analyses on intact bacteria are necessary to better understand the dynamics of C9 polymerisation.

Our prior studies have demonstrated that the presence of O-antigen correlates with potent C5 conversion and impaired C9 polymerisation in both *E. coli* and a Kpn O1-antigen strain [5,15]. Our data extends on these findings by showing that Kpn strains of different O-antigen types that resist MAC-dependent killing due to excessive C5 conversion require C9 supplementation to restore killing. This suggests this phenomenom may not be strictly dependent on O-antigen type. Future studies should investigate how the O-antigen causes an increase in C5 conversion and whether it is an antibody-mediated effect or driven by another mechanism.

Another key finding of this study is that excessive C5 conversion induced by an excess of mAbs paradoxically inhibited MAC-dependent killing. A prior study has also observed that an excessive amount of antibodies can induce a 'hook curve' effect in various ways, either by reducing complement deposition and killing in the presence of neutrophils [17], however MAC-dependent killing was not directly assessed. Our data addresses this gap and provides direct evidence that an excess of antibody induces potent C5 conversion which disrupts the C5b-to-C9 ratio, resulting in less MAC formation and subsequent bacterial killing.

Acknowledging that an excess of antibody can inhibit MAC-dependent killing is essential for the interpretation of antibody-dependent killing assays. It highlights that antibody titration is crucial to determine if an antibody has bactericidal potential, as absence of killing could be due to absent or excessive complement activation. Additionally, it has significant implications for therapeutic antibody development in bacterial infections. It highlights that efficacy of a complement-activating antibody depends on whether the bacterium responsible for the infection is not cleared because of complement evasion or excessive complement activation.

Prior studies have previously reported that high anti-O-antigen IgG2 and IgA titres derived from urosepsis patients correlate with the prevention of killing of the cognate *E. coli* isolate by normal human serum [36]. The authors explain these antibodies as inhibitory due to their ability to 'cloak' the entire surface of the bacterium, blocking access for the MAC to damage the bacterial membrane and resulting in resistance to serum killing. Our study shows that an excessive amount of complement-activating monoclonal IgG1's also results in inhibiting MAC-dependent killing, but by excessive C5 formation which inhibits C9 polymerisation. Therefore, while our study shows inhibition of bacterial killing using a different IgG isotype, monoclonal IgG1s compared to the patient-derived IgG2s, high anti-O IgG titres could result in a paradoxical loss of bacterial killing in fundamentally different ways. Our findings may also suggest an alternative explanation for the inhibitory effect of patient-derived IgG2 antibodies. If these antibodies are potent complement activators, which has been shown for IgG2 antibodies specifically against LPS [37], they may similarly drive excessive C5 formation, limit C9 polymerisation, and impair MAC-dependent killing. Future studies using patient-derived antibodies that show high IgG titres in the absence of bacterial killing could test whether partially inhibiting C5 conversion or adding C9 restores MAC-mediated killing.

Finally, our findings have clinical relevance, particularly in neonates, who have naturally low C9 levels [19]. A previous study has shown that C9 supplementation enhances MAC-dependent killing of *E. coli* in neonatal serum [18]. We extend these findings to Kpn, demonstrating that both C9 supplementation and specific antibodies are necessary for MAC-mediated killing in neonatal plasma in isolates that are MAC-susceptible in adult serum. This effect was observed across term and preterm neonates, supporting its relevance across gestational ages. Interestingly, one term neonatal plasma sample with a gestational age of 39 weeks showed efficient MAC-dependent killing upon C9 supplementation only, suggesting Kpn-specific antibodies were already present. This is line with previous studies showing that term neonates had higher Kpn-specific IgG levels compared to preterm neonates [38]. Considering these insights, C9 supplementation could be explored as a therapeutic adjunct in drug-resistant, Gram-negative neonatal sepsis, although potential inflammatory risks must be carefully evaluated.

In parallel, our data suggest that C5 inhibitors could serve as therapeutic agents by modulating complement activation, particularly in tissues with lower complement levels, such as the brain and cerebrospinal fluid [39,40]. Our findings suggest that in some cases, MAC-resistance can be overcome by partially inhibiting C5 conversion. While our study used a C5 inhibitor as a model rather than a direct therapeutic candidate, this raises broader questions about whether excessive C5 conversion is an immune evasion strategy or part of a regulated immune response aimed at recruiting neutrophils which have been shown to effectively kill Kpn [17]. Therapeutically, C5 inhibitors could prevent excessive inflammation and septic shock while creating a serum environment more conducive to MAC-dependent killing. However, inhibiting C5 conversion could also reduce C5a production, potentially impairing neutrophil recruitment and further compromising the immune response against Kpn.

There are some limitations to our study. We did not account for phagocytosis, which could influence bacterial clearance efficiency. Additionally, although bacterial concentrations were similar across all assays, we did not evaluate the effect of bacterial concentration on the C5b-to-C9 ratio, which is therapeutically important due to the impact of bacterial load on complement concentrations in serum or tissues. Future studies should address these factors to better understand the interplay between complement activation, MAC formation, and bacterial resistance. Additionally, based on our findings, it would be valuable to investigate the *in vivo* effects of supplementing C9 in neonates using preclinical animal models.

In conclusion, our study provides new mechanistic insights into how Kpn resists MAC-dependent killing despite potent complement activation. We demonstrate that excessive C5 conversion can limit C9 polymerisation, resulting in inefficient C9 polymerisation of MAC pores and reduced bacterial killing. These findings provide fundamental insights into distinguishing the difference between serum- and MAC-resistance and the interpretation of serum bactericidal assays. Additionally, these insights can be used to optimise mAb-based therapies against bacterial infections and consider C9 supplementation as a potential treatment strategy, particularly for patients with complement deficiencies.

## Materials and methods

### Ethics

This study adheres to all applicable ethical guidelines. The collection of human adult blood from healthy volunteers was approved by the Medical Ethics Committee of the University Medical Centre Utrecht (METC protocol 07–125/C, approved on March 1, 2010). For the collection of cord blood, expecting mothers were counselled for inclusion in the study before or after giving birth (deferred consent procedure). The Ethics Committee for Biobanking of the University Medical Center Utrecht (UMCU) approved the collection protocol (TCBio 21/223, approved on June 14th, 2021). All donors provided written informed consent following the principles of the Declaration of Helsinki.

### Bacterial strains

Kpn isolates were collected from routine diagnostics in the medical microbiology department in the University Medical Centre Utrecht, The Netherlands, kindly provided by Jelle Scharringa, Janetta Top and Ad Fluit. Kpn 209 [25] was kindly provided by Axel Janssen (University Medical Centre Utrecht, The Netherlands; University of Lausanne, Switzerland). A summary of strains used in this study is depicted in S1 Table. Antibiotic resistant genes, hypermucovicosity genes, and capsule and O-antigen typing of Kpn isolates was performed using whole-genome sequences and the Kleborate online tool (v3) [41–43]. The string test was performed by slowly pulling a single colony from an agar plate with a loop. A positive result is indicated by a visible viscous string of bacterial material extending ≥5 mm from the colony to the loop [44].

### Reagents, serum and plasma

Phosphate-buffered saline (PBS) was prepared in-house, unless specified otherwise. RPMI (ThermoFisher) supplemented with 0.05% human serum albumin (HSA, Sanquin), further referred to as RPMI buffer, was used in all experiments, unless otherwise stated. Normal human serum (NHS) was prepared as previously described [6]. Briefly, blood was allowed to clot and centrifuged to separate serum from the cellular fraction. Serum of 15–20 donors was pooled and stored at−80 °C. Heat-inactivated serum was obtained by incubating NHS at 56 °C for 30 min. Sera depleted of complement factors C5 and C9 (C5- and C9- depleted serum) were obtained from Complement Technology. Preadsorbed serum (dNHS) was prepared as previously described [25] using strains Kp209, Kp209_CSTR and KpO1_1 to remove bacteria specific antibodies without affecting complement activity. Briefly, ice cold NHS was incubated with bacteria to allow binding of strain-specific antibodies. Bacteria were pelleted and preadsorbed sera was collected. Three depletion rounds were performed followed by a filtration step. Neonatal cord blood plasma samples were collected at the obstetrics department of the UMCU. Cord blood was drawn from the umbilical vein into r-Hirudin tubes (Sarstedt) after either vaginal birth or caesarean section. Samples were processed and stored anonymously. Plasma was collected after centrifugation at 1000 × g for 10 min and immediately stored at − 80 °C. Details on gestational ages of plasma samples used in this study are depicted in S2 Table.

### Expression and fluorescent labelling of complement proteins, complement inhibitors and antibodies

C5 was expressed in Expi293F cells in-house as a His-tagged recombinant protein and purified using HisTrap column. UNbC3b-1 [45] (C3b-specific nanobody and alternative pathway C5 inhibitor), Eculizumab, and C1qNb75 (C1q-specific nanobody and classical pathway inhibitor) were expressed and purified using the same approach as C5. Fluorescently labelled C9 with Atto488 was generated by recombinantly expressing C9-LPETG-His with GGG-$N_3$ in Expi293F cells, followed by site-specific AF488 labelling at the C-terminus using sortagging. $C9_{TMH1-lock}$ was produced as described previously [5] and labelled using the same approach as C9. Cy5-labelled C9 and $C9_{TMH1-lock}$ was produced as described previously [5]. UNbC3B-1 was also fluorescently labelled using the same approach. OmCI was produced and purified as previously described [46]. Human anti-O2 UKpn2 was identified and produced as demonstrated previously [17].

## Bacterial culture

Kpn strains were cultured on Lysogeny broth (LB) 1.5% agar plates at 37 °C. Single colonies were picked and cultured overnight in LB medium at 37 °C while shaking (600 rpm). The following day, the bacteria were sub-cultured by diluting the overnight culture 1:100 in fresh medium and grown to OD600 = 0.4–0.6 at 37 °C while shaking. Bacteria were washed twice with RPMI buffer by centrifugation at 12000 $x\,g$ for 2 min and resuspended to the indicated bacterial concentration in RPMI buffer.

## IM-permeabilisation assay

Freshly cultured bacteria (final OD600 = 0.05; ~5 x $10^7$ bacteria/ml) were incubated with 1 µM SYTOX Green nucleic acid stain (Invitrogen) and the indicated serum/plasma at 37 °C in a FLUOstar Omega plate reader (BMG Labtech) for 90–120 minutes, indicated per experiment. Serum/plasma was supplemented with the indicated complement proteins, complement inhibitors and/or antibodies. For C5 and C9 excess, serum was supplemented with the respective protein at indicated concentrations. The physiological concentration of C5 and C9 in full serum are 359 nM and 845 nM, respectively. Complete C5 inhibition was achieved by combining 15 µg/ml Eculizumab and 6 µg/ml OmCI. SYTOX Green fluorescence was measured every 60 or 90 seconds using an excitation wavelength of 484 ± 15 nm, an emission wavelength of 527 ± 20 nm and a gain setting of 1000.

## Bacterial viability assay

After following a similar protocol as for the IM-permeabilisation assay, where freshly cultured bacteria (final OD600 = 0.05; ~5 x $10^7$ bacteria/ml) were incubated with the indicated serum/plasma at 37 °C for 90–120 minutes, indicated per experiment, serum reaction was stopped by dilution with PBS. First by diluting 100-fold in PBS followed by serial 10-fold dilutions in PBS that were spotted in duplicate on LB agar plates. Plates were incubated overnight at 37 °C and colony forming units (CFU/ml) were counted the next day to determine the bacterial concentration in CFU/ml.

## C3b deposition

Freshly cultured bacteria ($OD_{600}$ = 0.05; 5 x $10^7$ bacteria/ml) were incubated for 30 min at 37°C shaking (600 rpm) in 5% NHS (diluted in PBS) with 6 µg/ml OmCI and 15 µg/ml Eculizumab, HI-NHS as a control for the serum environment without functional complement proteins, or RPMI buffer. Bacteria were washed by centrifugation (3500 rpm for 5 min at 4°C) and resuspended in UNbC3b-1-AF488 to incubate for 30 min at 4°C shaking (600 rpm). Bacteria were washed again and fixed with 1% PFA to ~5 x $10^6$ bacteria/ml. Fluorescence was detected via flow cytometry (MACSQuantX, Miltenyi Biotech).

## C5a sandwich ELISA

After following a similar protocol as for the IM-permeabilisation assay, samples containing bacteria were pelleted by centrifugation, and the supernatant was harvested. Supernatants were diluted 10-, 15-, or 30-times and tested for C5a in a sandwich ELISA. Nunc Maxisorp ELISA plates were coated with 1 µg/ml of a C5a capture mouse mAb in PBS at 4°C overnight (R&D, DY2037). Plates were blocked with 4% BSA in PBS-Tween 0.05% (PBS-T) for one hour at 37°C, followed by incubation with the bacterial supernatant. C5a was detected using 0.2 µg/ml of primary C5a detection antibody which is specific for C5a and not native C5 (R&D, DY2037), followed by incubation with 1:5000 HRP-linked Streptavidin (Southern biotech, #7100–05). Plates were developed for 15 minutes with fresh tetramethylbenzidine (TMB) substrate solution and the reaction terminated with 2N sulfuric acid, after which OD450 was measured.

At each step for all ELISAs, 50 µl was added per well, antibodies were diluted in PBS-T + 1% BSA and incubation was done for 60 min at RT (except for coating or otherwise specified). Wells were washed three times with PBS-T between each step. C5a quantification was accomplished by interpolating from a standard curve of purified C5a (Bachem).

## C9 deposition and polymerisation

Freshly cultured bacteria ($OD_{600}$ = 0.05; 5 x $10^7$ bacteria/ml) were incubated with 5% C9-depleted human serum complemented with either directly labelled C9-Atto488 or $C9_{TMH1-lock}$-Atto488 at serological concentrations for 15 minutes at 37 °C, shaking (600rpm). Complement inhibitors OmCI, Eculizumab, C1qNb75 and UNbC3b-1 were separately titrated in sera reconstituted with $C9_{wt}$ and $C9_{TMH1-lock}$. RPMI buffer, HI C9-depleted serum reconstituted with C9wt or $C9_{TMH1-lock}$ and 5% C9-depleted serum were used as controls. Incubation was terminated by diluting and fixing the sample, 1:10 in ice-cold RPMI buffer + 1% PFA. Fluorescence was detected via flow cytometry (FACSVerse, BD Biosciences). C9 polymerisation ratio was calculated by dividing the median of C9-Atto488 by the median of $C9_{TMH1-lock}$-Atto488 of the gated bacterial population.

## Flow cytometry, data analysis and statistical testing

Flow cytometry data measured from FACSVerse (BD Biosciences) and MACSQuant X (Miltenyi Biotch) was analysed in FlowJo V.10 and bacteria were gated on FSC and SSC. Data were collected as the indicated number of biological replicates in the Fig legends and analysed using GraphPad Prism version 9.4.1 (458) and Microsoft Excel version 16.87. Statistical analyses are further specified in the Fig legends.

## Supporting information

**S1 Fig. Supplementing C9 in NHS increases MAC-dependent killing of multiple Kpn isolates. (a)** Bacterial viability at 90 min and **(b-h)** IM-permeabilisation overtime as determined by SYTOX fluorescence for multiple Kpn strains overtime until 90 min. The indicated Kpn strains (each ~5 × $10^7$ bacteria/ml) were incubated with buffer, 5% NHS supplemented with or without 560 nM C9 or MAC-inhibitors (6 µg/ml OmCI and 15 µg/ml Eculizumab) at 37°C for 90 min. Bacterial viability is depicted as log-transformed CFU/ml values. **(a)** Data represents mean ± SD of three independent experiments. **(b-i)** Representative IM-permeabilisation assay from three independent experiments.
(TIF)

**S2 Fig. Supplementing C9 in different concentrations of NHS increases MAC-dependent killing of multiple Kpn isolates. (a)** Bacterial viability at 90 min and **(b-c)** IM-permeabilisation overtime as determined by SYTOX fluorescence for Kp209. Kp209 (each ~5 × $10^7$ bacteria/ml) was incubated with buffer, NHS (10%, or 20%), supplemented with or without 560 nM C9 or MAC-inhibitors (6 µg/ml OmCI and 15 µg/ml Eculizumab) at 37°C for 90 min. Bacterial viability is depicted as log-transformed CFU/ml values. **(a)** MAC-inhibition control is indicated by the upper horizontal line. **(d)** Multiple Kpn strains were incubated with 5% NHS for 30 min at 37°C. C3b deposition was detected using UNbC3b-1-AF488 and analysed by flow cytometry by plotting histograms of the fluorescence intensity of bacterial populations. **(a)** Data represents mean ± SD of three independent experiments. **(b-c)** Representative IM-permeabilisation assay from three independent experiments. **(d)** Representative histograms from three independent experiments. Statistical analysis was performed for **(a)** using multiple paired t-tests; significant differences are displayed as *p < 0.05 and ***p < 0.001.
(TIF)

**S3 Fig. Partial inhibition of C5 conversion with C5 inhibitor increases MAC-dependent killing of KpO1_1.** KpO1_1 was incubated in 5% NHS and a concentration range of C5 inhibitor, OmCI, for 90 min at 37 °C. **(a)** Bacterial viability was determined after 90 min and depicted as log-transformed CFU/ml values. **(b)** IM-permeabilisation was depicted by plotting SYTOX fluorescence intensity after 90 min. **(c)** KpO1_1 was incubated for 90 min at 37 °C in 5% C9-depleted serum or repleted with 5% equivalent of either $C9_{wt}$-Atto488 or $C9_{TMH-1\ lock}$-Atto488 with a concentration range of OmCI or MAC-inhibitors (6 µg/ml OmCI and 15 µg/ml Eculizumab). Bacterial viability was determined after 90 min and depicted as log-transformed CFU/ml values. Horizontal dotted lines represent the level of the buffer control, the MAC-inhibition control

(3% dNHS with 6 µg/ml OmCI and 15µg/ml Eculizumab) or the assay's limit of detection (LoD), as indicated. **(a, b, c)** Data represents mean ± SD of three independent experiments. Statistical analysis was performed for **(a)** using a one-way ANOVA with Tukey's multiple comparisons' test where significant differences versus 5% NHS are indicated. A two-way ANOVA with Šídák's multiple comparisons' test was performed for **(c)** where all significant differences are indicated. Significant differences are indicated as **p≤0.01 and ****p < 0.0001.
(TIF)

**S4 Fig. Partial inhibition of C5 conversion with different complement inhibitors increases MAC-dependent killing on Kpn strain.** KpO1_1 (~$5 \times 10^7$ bacteria/ml) was incubated with 5% NHS supplemented with a concentration range of **(a)** Eculizumab, a C5 inhibitor, **(b, c)** UNbC3b-1, an alternative pathway C5 convertase inhibitor at 37 °C for 90 min or **(e)** C1qNb75, a C1q inhibitor. **(a, b, e)** IM-permeabilisation was depicted by plotting SYTOX fluorescence intensity after 90 min. **(c)** Bacterial viability was determined after 90 min and depicted as log-transformed CFU/ml values. **(d)** KpO1_1 was incubated for 90 min at 37 °C in either C9-depleted serum or its heat inactivated (HI) form repleted with 5% equivalent of either C9wt-Atto488 or $C9_{TMH-1\ lock}$-Atto488 with a concentration range of UNbC3b-1. Atto488 fluorescence was detected by flow cytometry. C9 polymerisation on KpO1_1 was calculated by dividing the median of the bacterial population incubated with $C9_{wt}$-Atto488 by the median of the bacterial population incubated with $C9_{TMH1-lock}$-Atto488. Horizontal dotted lines represent the level of the buffer control, the MAC-inhibition control (5% NHS with 6 µg/ml OmCI and 15µg/ml Eculizumab), heat-inactivated (HI)-C9 depleted serum or the assay's limit of detection (LoD), as indicated. Vertical dotted lines indicate UNbC3b-1 concentration that led to maximum killing. **(a, b, c, d, e)** Data represents mean ± SD of three independent experiments. Statistical analysis was performed for **(c)** using a one-way ANOVA with Tukey's multiple comparisons' test. Significant differences versus 5% NHS are indicated as ***p < 0.001.
(TIF)

**S1 Table. Bacterial strains used in this study.**
(DOCX)

**S2 Table. Neonatal plasma donors used in this study [47].**
(DOCX)

## Acknowledgments

The authors would like to acknowledge Carla J. C. de Haas for expressing the complement proteins, inhibitors and antibodies. The authors would like to thank Jelle Scharringa, Janetta Top and Ad Fluit for providing the Kpn strains used in this study. Additionally, the authors would like to thank the parents who consented to donating cord blood of their newborns for this study.

## Author contributions

**Conceptualization:** Kulsum M. Dawoodbhoy, Coco R. Beudeker, Bart W. Bardoel, Dennis J. Doorduijn.

**Formal analysis:** Kulsum M. Dawoodbhoy.

**Funding acquisition:** Michiel van der Flier, Suzan H. M. Rooijakkers.

**Investigation:** Kulsum M. Dawoodbhoy, Panagiotis Theofilidis, Frerich M. Masson, Dennis J. Doorduijn.

**Methodology:** Bart W. Bardoel, Dennis J. Doorduijn.

**Supervision:** Bart W. Bardoel, Dennis J. Doorduijn.

**Validation:** Kulsum M. Dawoodbhoy, Dennis J. Doorduijn.

**Visualization:** Kulsum M. Dawoodbhoy, Bart W. Bardoel, Dennis J. Doorduijn.

**Writing – original draft:** Kulsum M. Dawoodbhoy, Bart W. Bardoel, Dennis J. Doorduijn.

**Writing – review & editing:** Kulsum M. Dawoodbhoy, Coco R. Beudeker, Michiel van der Flier, Suzan H. M. Rooijakkers, Bart W. Bardoel, Dennis J. Doorduijn.

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
