## [Decision Letter · Decision Letter 0]

11 Mar 2026

PPATHOGENS-D-25-03185

Excessive C5 conversion prevents C9 polymerisation and subsequent MAC-dependent killing of Klebsiella pneumoniae

PLOS Pathogens

Dear Dr. Bardoel,

Thank you for submitting your manuscript to PLOS Pathogens. After careful consideration, we feel that it has merit but does not fully meet PLOS Pathogens's publication criteria as it currently stands. Therefore, we invite you to submit a revised version of the manuscript that addresses the points raised during the review process.

We look forward to receiving your revised manuscript.

Kind regards,

Leigh Knodler

Academic Editor

PLOS Pathogens

David Skurnik

Section Editor

PLOS Pathogens

Sumita Bhaduri-McIntosh

Editor-in-Chief

PLOS Pathogens

orcid.org/0000-0003-2946-9497

Michael Malim

Editor-in-Chief

PLOS Pathogens

orcid.org/0000-0002-7699-2064

**Journal Requirements:**

At this stage, the following Authors/Authors require contributions: Kulsum Mehboob Dawoodbhoy, Panagiotis Theofilidis, Frerich M. Masson, Michiel van der Flier, Suzan H. M. Rooijakkers, Bart W. Bardoel, and Dennis J. Doorduijn. Please ensure that the full contributions of each author are acknowledged in the "Add/Edit/Remove Authors" section of our submission form.

2) We have noticed that you have uploaded Supporting Information files, but you have not included a list of legends. Please add a full list of legends for your Supporting Information files after the references list.

3) We notice that your supplementary figures are uploaded with the file type 'Figure'. Please amend the file type to 'Supporting Information'. Please ensure that each Supporting Information file has a legend listed in the manuscript after the references list.

4) In the online submission form, you indicated that This paper does not report original code. Any additional information required to reanalyse the data reported in this paper is available from the lead contact upon request.. All PLOS journals now require all data underlying the findings described in their manuscript to be freely available to other researchers, either

1. In a public repository

2. Within the manuscript itself

3. Uploaded as supplementary information.

**Reviewers' Comments:**

Reviewer's Responses to Questions

**Part I - Summary**

Reviewer #1: This is a nicely done study assessing the roles of C5 cleavage, C9:C5 ratios, and antibody mediated C5 cleavage in MAC-dependent killing of Klebsiella pneumonia strains in vitro. I particularly think that the data on antibody titration is crucial to the field. The discussion is very well written but I had some challenges following the results and methods as described below. I have also a few general notes/comments for the authors.

Reviewer #2: The manuscript investigates the mechanism by which some Klebsiella pneumoniae strains are more resistant to MAC-dependent killing. They perform several experiments and find that excessive C5 conversion limits C9 polymerization and subsequent killing. Supplementing C9 restores pathogen killing. The results are particularly dramatic in sera from neonates, which are naturally C9-deficient and more susceptible to infection by Kpn and other microbes.

Overall, this is an important manuscript that sheds new light on mechanisms of complement activation and bacterial killing. The experiments are well conducted and overall, well interpreted. The discussion is balanced and addresses limitation of the work. I have a few minor comments below

**Part II – Major Issues: Key Experiments Required for Acceptance**

Reviewer #1: 1. Could the authors provide additional details on how the KP strains were selected? KP have significant genetic heterogeneity based on capsule composition that may influence MAC killing so additional details would be helpful re: generalizability. The authors have provided K type and O antigen type, which is helpful, but additional information that may also help the field would be sequence type, antibiotic resistance profiles (if these are clinical or multi-drug resistant), additional details on unpublished strains, clinical isolate versus laboratory strain, etc.

2. The strain nomenclature is challenging to follow - for example, what does Kp01_1 vs. Kp01_2 mean? Recommend simplifying the nomenclature or perhaps grouping by biology (e.g. by O antigen type). Also,  Table S1 should probably be included in the main text.

3. I have used 5% NHS in complement binding assays but not for killing assays - it appears the serum was allowed to kill overnight? If I understand their assay correctly, I would be interested to know if similar effects could be seen at shorter timepoints (e.g. 30 minutes) and higher serum concentrations that maybe more “physiologic.” I think higher concentrations are particularly important to test. If the authors are willing, I would perhaps target their efforts to only a key experiment or two?

Reviewer #2: 1. Figure and legends need to better clarify the number of samples used; this is not always evident, and in some figures, this is not sufficient. For example, in Figure 1b: it seems that some data points were performing with only an n=2, which is not sufficient. The experiments need to be repeated with an n=3 at the very minimum. Similarly, in Fig. 1c, it is unclear how many samples were used for each time point. This is true for all figures where the averages were shown.

**Part III – Minor Issues: Editorial and Data Presentation Modifications**

Reviewer #1: 1. Figure 2C - I am confused by this figure - I think this is another way to represent their findings in 2A. But if they are representing “killing” should it not be the initial inoculum - final yield for each strain and then measure that ratio for each strain? Rather than just calculating ratio of final yield?  If that is what the authors did, it was not immediately clear. Also, would a ratio <1 indicate impaired killing? That would be how I interpret a ratio, which is not consistent with 2A.

2. Minor point: I am not particularly concerned with statistical significance when the authors are showing multi-log differences in bacterial killing but in Figure 2a there seems to be minimal changes that influence their results. Would a non-parametric test like Mann-Whitney change statistical significance? This is a minor point.

3. Lines 466-468: Regarding neonatal supplementation of C9, would recommend testing in pre-clinical animal models to assess its effect in vivo.

4. For future submissions, It would be helpful to have figure legends co-located with the figures. Or at the end of the text rather than embedded in the text without the adjacent figure.

Reviewer #2: 2. The strains in figure 2 belong to many O and K antigens. Given that capsules are important for complement resistance, can the authors include any information on the K antigens of the strains? Are some of the strains hypermucoviscous? Some appear to be quite resistant to killing.

3. In Fig. 4D, it seems that killing is not restored to baseline levels without C9 even when IgG1 UKpn2 is above 0.1ug/ml. Please clarify and/or revise the data interpretation

PLOS authors have the option to publish the peer review history of their article (what does this mean?). If published, this will include your full peer review and any attached files.

**Do you want your identity to be public for this peer review?** For information about this choice, including consent withdrawal, please see our Privacy Policy.

Reviewer #1: No

Reviewer #2: No

**Figure resubmission:**
---

## [Editor Report · Decision Letter 1]

24 Apr 2026

Dear Dr Bardoel,

We are pleased to inform you that your manuscript 'Excessive C5 conversion prevents C9 polymerisation and subsequent MAC-dependent killing of Klebsiella pneumoniae' has been provisionally accepted for publication in PLOS Pathogens.

Best regards,

Leigh Knodler

Academic Editor

PLOS Pathogens

David Skurnik

Section Editor

PLOS Pathogens

Sumita Bhaduri-McIntosh

Editor-in-Chief

PLOS Pathogens

orcid.org/0000-0003-2946-9497

Michael Malim

Editor-in-Chief

PLOS Pathogens

orcid.org/0000-0002-7699-2064
---

## [Editor Report · Acceptance letter]

Dear Dr Bardoel,

We are delighted to inform you that your manuscript, "Excessive C5 conversion prevents C9 polymerisation and subsequent MAC-dependent killing of Klebsiella pneumoniae," has been formally accepted for publication in PLOS Pathogens.

Best regards,

Sumita Bhaduri-McIntosh

Editor-in-Chief

PLOS Pathogens

orcid.org/0000-0003-2946-9497

Michael Malim

Editor-in-Chief

PLOS Pathogens

orcid.org/0000-0002-7699-2064